# Deoxynivalenol in the Diet Impairs Bone Mineralization in Broiler Chickens

**DOI:** 10.3390/toxins11060352

**Published:** 2019-06-18

**Authors:** Marsel Keçi, Annegret Lucke, Peter Paulsen, Qendrim Zebeli, Josef Böhm, Barbara U. Metzler-Zebeli

**Affiliations:** 1Institute of Animal Nutrition and Functional Plant Compounds, Department for Farm Animals and Veterinary Public Health, University of Veterinary Medicine Vienna, Veterinärplatz 1, 1210 Vienna, Austria; Marsel.Keci@vetmeduni.ac.at or mkeci@ubt.edu.al (M.K.); annegret.lucke@googlemail.com (A.L.); Qendrim.Zebeli@vetmeduni.ac.at (Q.Z.); 2Department of Animal Production, Faculty of Agriculture and Environment, Agricultural University of Tirana, Kodër Kamëz, SH1, 1000 Tirana, Albania; 3Institute of Meat Hygiene, Department for Farm Animals and Veterinary Public Health, University of Veterinary Medicine Vienna, Veterinärplatz 1, 1210 Vienna, Austria; Peter.Paulsen@vetmeduni.ac.at

**Keywords:** broiler chicken, deoxynivalenol, bone mineralization, calcium, phosphorus

## Abstract

Deoxynivalenol (DON) is one of the most abundant and important trichothecene mycotoxins produced by *Fusarium* species. In chickens, DON intake causes feed refusal, impairs performance, gut barrier function, and immunity, and raises oxidative stress. To determine the effect of DON on bone mineralization and serum calcium and phosphorus, 80 newly-hatched chickens were fed 4 diets with 0, 2.5, 5, and 10 mg DON/kg feed in this pilot study. In week 5, chickens were euthanized, femur and tibiotarsus bones were separated from the meat, and after incineration ash composition, as well as serum calcium and phosphorus, were determined using clinical biochemistry. Dietary DON reduced chicken dry matter, calcium, and phosphorus intake, and subsequently body and leg weight. DON affected bone density and composition of the tibiotarsus more drastically than of the femur. However, lower mineral intake did not solely explain our observations of the quadratically lower tibiotarsus density and ash content, as well as linearly decreased Ca content in the femur and tibiotarsus with increasing DON levels. Linearly decreasing serum phosphorus concentrations with increasing DON levels further supported impaired mineral homeostasis due to DON. In conclusion, already low dietary DON contamination of 2.5 mg/kg feed can compromise bone mineralization in chickens.

## 1. Introduction

Deoxynivalenol (DON) is one of the most abundant and important trichothecenes, mainly produced by *Fusarium graminearum* and *F. culmorum* [1]. Chemically, trichothecenes belong to the group of sesquiterpenoids containing the 12, 13 epoxide group, considered to be critical for their toxicity and sharing a 9, 10 double bond [2]. *Fusarium* species infect important grains in agriculture, especially in the northern temperate regions. It is considered as the most widespread mycotoxin producing genus found in cereals that are grown in Europe, Asia, and America [3,4]. Due to its resistance to decontamination processes, such as milling, processing, and heating, DON readily enters animal feed [5].

The general toxic effects of prolonged exposure to DON in experimental animals, such as mice, rats, dogs, cats, and swine, are anorexia, decreased weight gain, and altered nutritional efficiency [6], whereas in broiler chickens different adverse effects of DON were reported in the literature. Recently, we found reduced body weight, body weight gain, and dry matter intake in broiler chickens when fed feed contaminated with increasing DON concentrations (2.5–10 mg/kg) [7,8]. Moreover, we could show in this chicken model that the increasing dietary DON concentrations upregulated the innate immune response in the small intestine, as indicated by elevated expression levels of toll-like receptors-2, interleukin-6, and claudin-1 in the duodenal mucosa, and claudin-1, toll-like receptors-2, and transforming growth factor-β 1 expression in the jejunal mucosa [7,8].

Although leg problems are mostly due to genetic selection for growth [9,10,11], mycotoxins as ubiquitous contaminants may further contribute to poor leg health and bone mineralization in modern fast-growing chickens. Against this background, *Fusarium* toxins, such as T-2, and other mycotoxins, such as ochratoxin and aflatoxin, were reported to decrease the breaking strength, mineral content, percentage of calcium (Ca) and phosphorus (P) in tibiotarsus, and increase manifestation of lameness, leg weakness, and tibial dyschondroplasia in chickens [12,13,14]. However, this has been poorly studied in regards to DON contamination in feed for broiler chickens so far. Serum mineral levels have been used, along with other parameters, as indicators for bone health, and should therefore reflect alterations in the mineralization status of the bones potentially caused by dietary DON.

The aim of this pilot study was to evaluate whether increasing concentrations of DON in the diet affect leg composition, as well as physical (weight, volume, density) and gravimetric (dry matter, ash content, and ash density) characteristics of femur and tibiotarsus. Other parameters that were evaluated were Ca and P concentration in serum, content of Ca and P, and Ca:P ratio in femur and tibiotarsus ash. We hypothesized that DON may impair leg bone strength due to its effect on bone mineralization. We also hypothesized that serum Ca and P concentrations would correspond to the mineralization status of the bones, as affected by increasing dietary DON contamination.

## 2. Results

Chickens were fed diets that were contaminated with increasing levels of DON. The dry matter intake (DMI) remained similar for all DON levels in the first two weeks of life (Table 1), and only started to decline from the third week of life in the birds fed the DON-contaminated diets (*p* < 0.05), irrespective of the DON level. Whilst this negative effect of DON on DMI amounted to 3.5% from day 14 to 21, it became stronger until the fifth week of life (*p* < 0.05), representing a decline in DMI of 8.9 and 12.6% for days 22–28 and 29–33, respectively. This caused an overall reduction in DMI of 7.8% for the whole experimental period in birds exposed to the dietary DON compared to those fed the 0 mg/kg DON diet. Despite the decreasing DMI in DON-fed chickens, the linear increase in the intake of DON reflected the increasing dietary concentrations, being about three- and five-times the amount in the 5 and 10 mg/kg DON groups compared to the 0 mg/kg DON group throughout the experimental period (*p* < 0.001). Due to the lower DMI intake from the third week of life, the Ca and P intake also decreased in the chickens receiving the DON-contaminated diets. The decline in Ca and P intake amounted to 7.8 and 7.7%, respectively, for the whole experimental period in chickens fed the DON-contaminated diets compared to birds fed the 0 mg/kg DON diet. Concurrently, the body weight (BW) gain responded differently to the increasing DON concentrations throughout the experimental period, showing trends for lower BW gain for days 1–7, 15–21, and 29–33 when comparing the chicken groups that received contaminated versus non-contaminated feed (*p* < 0.10). When calculated for the whole experimental period, DON decreased the BW gain by 6.0% (*p* < 0.05).

In order to assess the effect of increasing DON concentrations on the leg composition, legs were thawed. The weight of the thawed leg decreased linearly (*p* = 0.05) with increasing dietary DON concentrations (Table 2). Similarly, the meat weight of the thawed leg decreased linearly (*p* = 0.039) with increasing DON concentrations.

For all measurements taken on the leg bone, only the tibiotarsus density quadratically decreased with increasing DON concentrations, maximally declining by 2.0% in the 2.5 mg/kg DON group, (Table 3).

DON decreased the ash content in dry matter (AsC_(DM)_) of the femur (*p* < 0.05) and tibiotarsus (*p* < 0.10) (Table 4), whereby the effect was strongest in the 2.5 mg/kg DON group. Likewise, the ash density (Asd) of the tibiotarsus quadratically responded to the increasing DON concentrations, with a maximal decrease of 10% in the 2.5 mg/kg DON group.

The increasing DON concentrations in the diet linearly decreased the serum P concentration (*p* = 0.034), whereas serum Ca was not affected by DON (Table 5). In both femur and tibiotarsus, Ca concentration in ash content linearly decreased with increasing DON concentrations by up to 1.9 and 3.8%, respectively (*p* < 0.001) (Table 5). This resulted in a linearly decreasing Ca:P ratio in both bones with increasing DON concentrations.

Pearson’s correlation analysis was used to characterize relationships of bone parameters and serum Ca and P with body weight (BW), daily feed DM intake (DMI), daily Ca and P intake, and daily DON intake (Table 6). The weight of the whole leg, as well as the weight and volume of femur and tibiotarsus positively correlated with BW across all chicken groups (*r* ≥ 0.58; *p* < 0.05), whereby the correlations were stronger between BW and femur weight (*r* ≥ 0.70) than between BW and tibiotarsus weight (*r* = 0.61). Similar positive correlations of daily DMI, Ca, and P intake with leg weight, femur weight, and volume existed (*r* ≥ 0.36) but these were weaker, whereas no correlations between daily DMI, Ca, and P intake and tibiotarsus parameters were found (*r* < 0.3). With regards to the Ca and P concentration in femur, tibiotarsus, and serum, only the Ca content in femur and serum P weakly correlated (*r* ≥ 0.30) to BW, DMI, and daily Ca intake, as well as femur Ca with daily P intake. The daily DON intake of the chickens correlated negatively with the Ca content in femur and tibiotarsus (*r* ≥ 0.35).

## 3. Discussion

Only little information was available from the literature on whether DON may contribute to leg problems in broiler chickens. For this reason, the results of the present pilot study show that feeding contaminated diets with graded levels of DON modified bone mineralization in 5-week-old chickens, emphasizing that DON can contribute to bone-related diseases in poultry production. By using classical evaluation techniques to study the bone composition, we therefore provide valid information for DON-related changes in femur and tibiotarsal bones. According to our data, especially the tibiotarsal bone density and the Ca reposition in both femur and tibiotarsus were affected by DON; the latter being supported by the present Pearson’s correlations. Results further demonstrate that serum Ca and P were not useful markers to indicate DON-related changes in bone mineralization. However, the decline in serum P with increasing DON concentrations suggested that besides Ca, the P homeostasis was also altered by DON. 

A decreased feed intake is a common symptom in birds exposed to dietary DON [15,16,17]. Contrary to our expectations, however, we did not see a clear linear effect of the increasing DON concentrations on daily DMI, indicating that the presence of DON at a level below the recommended upper limit for chicken feed may already critically suppress appetite and normal eating behavior. Albeit the chickens in the present study were exposed to dietary DON from their first day of life, their DMI was only depressed from the third week of life. This may suggest that a certain threshold needed to be reached above which DON exerted its detrimental effect on feed intake. The reduced BW gain and leg weight of chickens exposed to DON was then the consequence of the lower feed intake, as supported by the present positive correlations. For this reason, the reduced accretion of muscle tissue on thigh and drumstick clearly reflects the lower daily protein intake. Aside from the detrimental effect of DON on feed intake, DON has been reported to impair intestinal absorption and metabolism of amino acids, as well as its inhibitory effect on protein in general may have additionally limited muscle protein synthesis [15,16,17]. Although not having been measured specifically, other proteinogenous tissues, such as joint tissues and skin, were negatively affected by DON. 

As a consequence of the reduced DMI, DON depressed the daily intake of Ca and P from the third week of life, thereby decreasing the provision of the major micronutrients needed for ossification of the developing skeleton. As supported by the present correlations, it seems that the lower Ca and P intake of birds exposed to DON was more critical for the mineralization of the femur than for the tibiotarsus. Similar to muscle protein accretion, without having studied the pathways in the present experiment, DON-related alterations in gastrointestinal absorption or metabolism of Ca and P in bones and kidney may have contributed to the altered mineralization of the bones [18,19]. Furthermore, the cytotoxicity of DON may have directly affected the bone tissue structure, reducing the Ca content and affecting the gene expression of proteins involved in the bone mineralization process [20]. Moreover, the ability of DON to enhance the apoptotic activity of osteoblasts, as indicated in a murine model [21], may also be a possible underlying mechanism, explaining the negative effect of DON on bone mineralization in the present study. In comparing the gravimetric characteristics and chemical composition of the tibiotarsus and femur, it became obvious that the mineralization of the longer tibiotarsal bone was more compromised than that of the shorter femur by the increasing DON concentrations in the 5-week old chickens. Interestingly, we found quadratic effects for the tibiotarsal AsC_(DM)_ and Asd, as well as for the AsC_(DM)_ in the femur with increasing DON concentrations, showing that 2.5 and 5 mg DON/kg feed had the most detrimental effects. These effects are difficult to explain but cannot be solely associated with the lower DMI of the chickens fed the diets with the higher DON levels. In fact, the lower DMI intake in birds fed the diets with the 5 and 10 mg DON/kg feed did not have large consequences for the total amount of DON consumed on a daily basis, which was three- to five-times higher than in birds receiving the diet with 2.5 mg DON/kg feed. Therefore, other coping mechanisms may have taken place. According to the present data, mainly the Ca reposition in the femur and tibiotarsus was impaired, whereas the P content in these bones did not differ between treatments. In bones, the largest proportion of Ca and P is immobilized as insoluble hydroxyapatite crystals, i.e., Ca_10_(PO_4_)_6_(OH)_2_ [22], whereby single Ca-ions may be replaced by other divalent cations or may be missing [23]. Since the Ca and P intake was only lower from the third week of life, and in considering that the period of the highest bone mineralization and growth of the skeleton in modern chicken lines lays between days 7 and 28 of age [24,25,26], this may hint at the critical time point when DON most significantly impaired the formation of the bones, either directly by affecting the physiological processes or indirectly via reducing the birds’ DMI. This should be examined in more detail in future studies.

The femur is often used as a representative bone for the evaluation of bone mineralization for the whole body skeleton [27]. The present results point in the direction that it may be worth examining DON effects on other bones, such as the tibiotarsus, as effects may differ. Therefore, it may be valid to more closely investigate DON effects on the ossification of vertebrae and rip bones in future studies to fully understand the impact of mycotoxins, such as DON, on bone mineralization. The present results let speculate that peripheral bones were less well-supplied with the necessary nutrients than more central ones, which may be supported by the stronger linear decline in the Ca concentration of the tibiotarsus compared to that of the femur with increasing DON concentrations. This would also be supported by the present correlation analysis, as Pearson’s correlation coefficient was more negative between DON intake and the tibiotarsus Ca content than between DON intake and femur Ca.

It is interesting to note that serum Ca levels remained similar across all DON concentrations, whereas serum P linearly increased with increasing DON contamination, corresponding to the lower P intake. This may be linked to the many important physiological roles of Ca in the body (e.g., nerve communication, second messenger activity, enzymatic cofactor activity, hormonal secretion, and blood coagulation), making it essential to keep serum Ca levels in a narrow range [28,29]. From the present data, it may be deduced that chickens may have compensated for a potential drop in serum Ca levels, thereby decreasing the utilization of Ca for bone formation. As serum P levels are not as tightly controlled as serum Ca [18], this may have led to a normal built-in of phosphate ions into the bones despite a potentially lower intestinal absorption or greater renal excretion, leading to the linear decrease in serum P with increasing DON concentrations. However, the missing correlations between most bone parameters and serum Ca and P highlights that both minerals in serum are not very useful for predicting alterations of bone mineralization in fast-growing chickens.

In conclusion, our findings show that feeding DON-contaminated diets, even at a dietary level of 2.5 mg/kg, reduced serum P and bone mineralization, especially the Ca assimilation, in 5-week-old broiler chickens. More specifically, increasing DON levels affected the bone density, ash content, and ash density in femur and tibiotarsus, inducing a stronger negative effect on the tibiotarsus bone than on the femur. In addition, DON reduced the yield of saleable meat (i.e., thigh and drumstick) and increased the drip losses during the thawing process. Although part of the observed DON effects can be linked to the DON-associated reduced DMI and mineral intake, results also hint at other underlying physiological mechanisms, possibly at the molecular level, which we have not assessed in this pilot study. Therefore, further research is warranted to investigate the molecular basis of the effect of DON on mineral homeostasis and bone formation at histo-morphological and gene and protein expression levels. Taking into account the key role of bone mineralization for bone strength and overall health and welfare, the present data suggest the negative impact of DON on broiler performance and welfare.

## 4. Materials and Methods 

### 4.1. Ethics Statement

The animal experiment was approved by the institutional ethics and animal welfare committee and the national authority according to paragraph 26 of Animal Experiments Act, Tierversuchsgesetz 2012-TVG 2012 on the 1 April 2015 (BMWFW-68.205/0062-WF/V/3b/2015).

### 4.2. Animals and Experimental Design

The experimental design, including housing conditions, diets, and performance parameters, were described previously [7,30]. In brief, a total of 80 one-day-old broiler chicks (ROSS 308) were randomly assigned to 4 feeding groups, which received diets with increasing concentrations of DON. All groups were fed ad libitum the same basal diet, which was contaminated with increasing DON concentrations (Romer Labs, Tulln, Austria): control group without artificial DON contamination, or 2.5 mg/kg DON group, 5 mg/kg DON group, and 10 mg/kg DON group, who were experimentally contaminated with 2.5, 5, and 10 mg/kg of DON, respectively. The basal diet consisted of 57.59% wheat, 25.44% soybean meal, 13.00% protein-vitamin-mineral supplement, 2.97% rapeseed oil, and 1.00% megafat [7]. Diet samples were analyzed for proximate nutrients and Ca and P content, as recently described [7]. The analyzed chemical composition of the diet is shown in Table 7. Before euthanization between days 34 and 37 of life, feed was withdrawn for 0.5–1.5 h. The chickens were euthanized by an overdose of thiopental (50–100 mg/kg BW; Medicamentum Pharma GmbH, 8643 Allerheiligen im Mürztal, Austria) into the wing vein followed by exsanguination.

### 4.3. Bone Preparation and Bone Measurements

Both legs were separated from the carcass at the hip joint (articulatio coxae). The foot (tarsometatarsus and toes) was removed. The remaining parts of the leg, the proximal (thigh) and the distal part (drumstick), were separated by cutting the hock joint (intertasel joint) and stored at −20 °C until further analysis. In order to account for drip losses during the thawing process [31], the frozen weight of the leg was determined before thawing the leg overnight at room temperature. The next day, the leg was dried using paper towels and weighed again [32]. The drip loss was calculated by the following formula:(1)$$Drip\;loss\;\left(\%\right)=\left(1-\frac{ThW}{FzW}\right)*100$$
where *ThW* is the thawed weight and *FzW* is the frozen weight. 

The meat was manually dissected from the femur and tibiotarsus, and all three samples (meat, femur, and tibiotarsus) were weighed separately. The volume of tibiotarsus and femur was calculated by determining the volume of distilled water displaced when the bone was immersed in a 100 mL graduated cylinder. The measurement was repeated three times. The bone density (BoDe, g/mL) was calculated by dividing the fresh bone weight (BoW) in g by the volume (V) in mL,
(2)$$BoDe=\frac{BoW}{V}.$$
The bones were frozen at −20 °C until further analysis.

### 4.4. Determination of Dry Matter, Ash Content and Ash Density.

The bones were again defrosted overnight and cut into three different parts to accelerate the drying process. The samples were dried for 22 h at 110 °C in a drying oven (Memmert U40 dry oven, Schwabach, Germany). The dry matter (DM) was calculated by dividing the dry weight (DW) with the fresh weight (FW),
(3)$$DM\;\left(\%\right)=\frac{DW}{FW}*100.$$

Bones were incinerated for 24 h at 600 °C in a muffle oven (carbolite CWF 1100, Carbolite Gero, Neuhausen, Germany). To avoid flame and smoke formation, during the incineration process inside the muffle oven, tibiotarsus and femur were firstly burned inside crucibles heated by a Bunsen burner. The ash weight (AsW) was measured and the AsC_(DM)_ and the ash content in fresh AsC_(FM)_ were calculated by the following formulas:(4)$$AsC_{\left(DM\right)}\left(\%\right)=\frac{AsW}{DM}*100$$
(5)$$AsC_{\left(FM\right)}\left(\%\right)=\frac{AsW}{FM}*100.$$

The ash density (Asd) was calculated by dividing the AsW (in g) of the bones with their respective volumes (V) in mL [33],
(6)$$Asd=\frac{AsW}{V}\;[33].$$

After the incineration, the bones were reduced to powder using mortar and pestle and stored in 2 mL Eppendorf tubes.

### 4.5. Calcium and Phosphorus Measurements

To measure the concentration of Ca (method 10.3.2) and P (method 10.6.1) [34] in the bones, approximately 100 mg of ash were inserted in a 25 mL volumetric flask and 5 mL of concentrated nitric acid (>65%) were added twice and shaken vigorously. The samples were left to react overnight. The next day, each sample solution was filled up to 25 mL with double-distilled water. The solution was filtered (Whatman cat no 1002 110 filter paper, GE Healthcare UK Limited, Little Chalfont, UK) into 15 mL graduated plastic tubes (Sarstedt, Nümbrecht, Germany). For the P measurement, the filtered sample solution was diluted by 100-times. The calibration curve was prepared using a 1:20-dilution series of the 1g/L P standard (0.4387 g KH_2_PO_4_/100 mL ddH_2_O), resulting in the following concentrations: 0, 0.5, 1, 2.5, 5, 10, 50, 30, 40, and 50 g/L. One mL of P containing solutions (calibration curve solution and filtered sample solution) was mixed with 1 mL of P colour reagent (vanadate–molybdate reagent) and 3 mL of double-distilled water. After 50 min, the absorbance was measured on an ultraviolet-visible spectrophotometer (model UV-1800 240V IVDD, Shimadzu Europa GmbH, Duisburg, Germany) at 405 nm wavelength.

The P concentration [P] was calculated according to the equation:(7)$${\left[P\right]}_{\left(\frac{g}{kg}\right)}=\frac{\frac{SolV}{AsW}*dil.Fac.*Conc.P}{1000}$$
where SolV is the final volume of solution in which the ash was dissolved (25 mL), AsW is the dissolved ash weight (g) in the flask, dil.Fac. is the dilution factor, and conc. P is the concentration of the P calculated from the calibration curve.

The Ca was analyzed by flame atomic absorption spectrometry (Perkin Elmer UV-VIS, model 4100, Perkin Elmer, Shelton, QT, USA) [34]. Calibration curve solutions and filtered sample solutions were prepared with 2% nitric acid (HNO_3_) solution. Modified Schinkel solution (CsCl, La(NO_3_)_2_
∗ 6H_2_O) was used as an interference error reductor. The absorbance of the samples was measured at 422.7 nm wavelength. For the calibration curve, three increasing Ca concentrations were prepared from the Ca standard solution dissolved in 2% HNO_3_ (PerkinElmer, Shelton, QT, USA), resulting in 1, 3, and 5 mg/L of Ca, respectively. For all dilution steps, double-distilled water was used.

Calcium concentration [Ca] in the ash was calculated according to the formula
(8)$${\left[Ca\right]}_{\left(\frac{g}{kg}\right)}=\frac{\frac{SolV}{AsW}*dil.Fac.*Conc.\;\;Ca}{1000}$$
where SolV is the final volume of solution in which the ash was dissolved (25 mL), AsW is the dissolved ash weight (g) in the flask, dil.Fac. is the dilution factor, and conc. Ca is the concentration of the Ca calculated using the calibration curve.

Serum Ca and P were determined by standard enzymatic colorimetric analysis using an autoanalyzer for clinical chemistry (Cobas 6000/c501; Roche Diagnostics GmbH, Vienna, Austria).

### 4.6. Statistical Analysis

For statistical analyses, the MIXED procedure of SAS (Version 9.4; SAS Institute Inc., Cary, NC, USA) was used. The experimental unit was defined as chicken with the covariance structure being variance compounds. The experimental run was included in the model as random effect. Linear and quadratic contrasts between control feeding and the DON-contaminated feeding groups were calculated using the CONTRAST statement of SAS. Degrees of freedom were estimated using the Kenward-Rogers method. Results are presented as least square means ± standard error mean (SEM). Pairwise comparisons between groups were performed using the pdiff option in SAS. Significance was declared at *p*-value (*p*) ≤ 0.05 and a tendency was stated if *p* ≤ 0.10.

To characterize the relationships between daily DMI, Ca, P, and DON intake, leg composition, physical characteristics of bones, bone mineralization, Ca and P content in bones, and serum Ca and P, Pearson’s correlation analysis was performed using CORR procedure of SAS.

## Figures and Tables

**Table 1 toxins-11-00352-t001:** Dry matter, calcium (Ca), phosphorus (P), and deoxynivalenol (DON) intake and body weight gain during every week of life and during the whole life of broiler chickens fed diets with increasing levels of DON (0, 2.5, 5, or 10 mg DON/kg diet). ^1,2^

Item	DON, mg/kg Feed				Contrast, *p* *		
	0	2.5	5	10	0 vs. DON	Linear	Quadratic
**Dry Matter Intake (g)**							
Day 1–7	153 ± 5.4	147 ± 5.4	148 ± 5.4	152 ± 5.4	0.535	0.942	0.376
Day 8–14	384 ± 21.3	354 ± 21.3	357 ± 21.3	379 ± 21.3	0.398	0.902	0.221
Day 15–21	549 ± 7.5	529 ± 7.5	529 ± 7.5	531 ± 7.5	0.030	0.128	0.135
Day 22–28	766 ± 17.8	693 ± 17.8	689 ± 17.8	711 ± 17.8	0.002	0.039	0.010
Day 29–33	656 ± 16.3	596 ± 16.3	553 ± 16.3	572 ± 16.3	<0.001	<0.001	0.015
Day 1–33	2481 ± 49.5	2293 ± 49.5	2251 ± 49.5	2320 ± 49.5	0.001	0.021	0.011
**DON Intake (mg)**							
Day 1–7	0.03 ± 0.03	0.30 ± 0.03	0.83 ± 0.03	1.60 ± 0.03	<0.001	<0.001	<0.001
Day 8–14	0.07 ± 0.12	0.72 ± 0.12	1.99 ± 0.12	3.99 ± 0.12	<0.001	<0.001	<0.001
Day 15–21	0.10 ± 0.05	1.08 ± 0.05	2.94 ± 0.05	5.59 ± 0.05	<0.001	<0.001	<0.001
Day 22–28	0.14 ± 0.10	1.41 ± 0.10	3.83 ± 0.10	7.48 ± 0.10	<0.001	<0.001	<0.001
Day 29–33	0.12 ± 0.11	1.21 ± 0.11	3.07 ± 0.11	6.01 ± 0.11	<0.001	<0.001	<0.001
Day 1–33	0.41 ± 0.41	4.31 ± 0.41	11.57 ± 0.41	22.53 ± 0.41	<0.001	<0.001	<0.001
**Ca Intake (g)**							
Day 1–7	2.51 ± 0.09	2.41 ± 0.09	2.43 ± 0.09	2.49 ± 0.09	0.544	0.939	0.389
Day 8–14	6.29 ± 0.35	5.79 ± 0.35	5.85 ± 0.35	6.21 ± 0.35	0.398	0.901	0.222
Day 15–21	8.99 ± 0.12	8.66 ± 0.12	8.67 ± 0.12	8.71 ± 0.12	0.030	0.131	0.128
Day 22–28	12.55 ± 0.29	11.35 ± 0.29	11.29 ± 0.29	11.65 ± 0.29	0.001	0.038	0.010
Day 29–33	10.75 ± 0.26	9.75 ± 0.26	9.05 ± 0.26	9.37 ± 0.26	<0.001	<0.001	0.015
Day 1–33	40.63 ± 0.81	37.55 ± 0.81	36.88 ± 0.81	38.01 ± 0.81	0.001	0.022	0.011
**P Intake (g)**							
Day 1–7	1.26 ± 0.04	1.22 ± 0.04	1.23 ± 0.04	1.26 ± 0.04	0.566	0.972	0.392
Day 8–14	3.17 ± 0.18	2.92 ± 0.18	2.95 ± 0.18	3.13 ± 0.18	0.401	0.906	0.222
Day 15–21	4.53 ± 0.06	4.37 ± 0.06	4.37 ± 0.06	4.39 ± 0.06	0.030	0.131	0.132
Day 22–28	6.33 ± 0.15	5.72 ± 0.15	5.69 ± 0.15	5.87 ± 0.15	0.001	0.038	0.010
Day 29–33	5.42 ± 0.13	4.91 ± 0.13	4.56 ± 0.13	4.72 ± 0.13	<0.001	<0.001	0.015
Day 1–33	20.4 ± 0.41	18.87 ± 0.41	18.53 ± 0.41	19.09 ± 0.41	0.001	0.022	0.012
**Body Weight Gain (g)**							
Day 1–7	99 ± 3.6	88 ± 3.7	94 ± 3.6	92 ± 3.6	0.077	0.391	0.338
Day 8–14	252 ± 9.31	241 ± 9.3	244 ± 9.3	256 ± 9.3	0.631	0.733	0.224
Day 15–21	400 ± 10.2	372 ± 10.5	382 ± 10.2	380 ± 10.2	0.068	0.270	0.236
Day 22–28	427 ± 16.1	443 ± 16.1	426 ± 16.1	459 ± 16.1	0.113	0.422	0.057
Day 29–33	353 ± 21.7	328 ± 21.7	325 ± 21.7	316 ± 21.7	0.098	0.107	0.589
Day 1–33	1576 ± 38.7	1470 ± 40.8	1472 ± 38.7	1503 ± 38.7	0.038	0.211	0.085

^1^ Values are least square means (LSM) ± standard error of the mean (SEM); *n* = 20. ^2^ Data on DMI and body weight gain for the whole experimental period were already presented in Lucke et al. [7]; ** p*-values for orthogonal contrasts to test linear and quadratic relationships between control feeding and increasing levels of DON, as well as the overall difference of 0 DON versus all DON groups (0 vs. DON).

**Table 2 toxins-11-00352-t002:** Leg composition and drip loss of 5-week old broiler chickens fed diets with increasing levels of deoxynivalenol (DON; 0, 2.5, 5, or 10 mg DON/kg diet). ^1^

Item	DON, mg/kg Feed				Contrast, *p* *		
	0	2.5	5	10	0 vs. DON	Linear	Quadratic
Drip loss (%)	2.7 ± 0.3	3.2 ± 0.3	3.3 ± 0.3	2.5 ± 0.3	0.373	0.751	0.035
Leg (g)	161 ± 4.5	153 ± 4.5	147 ± 4.6	152 ± 4.5	0.050	0.116	0.141
Meat (g)	134 ± 3.9	127 ± 3.9	121 ± 4.0	126 ± 3.9	0.039	0.089	0.133

^1^ Values are least square means (LSM) ± standard error of the mean (SEM); *n* = 16 in all groups for the drip loss, *n* = 19 in the 5 mg/kg DON group had, and *n* = 20 in the remaining groups for the other parameters; * *p*-values for orthogonal contrasts to test linear and quadratic relationships between control feeding and increasing levels of DON, as well as the overall difference of 0 DON versus all DON groups (0 vs. DON).

**Table 3 toxins-11-00352-t003:** Physical characteristics of femur and tibiotarsus in 5-week-old broiler chickens fed diets with increasing levels of deoxynivalenol (DON; 0, 2.5, 5, or 10 mg DON/kg diet). ^1^

Item	DON, mg/kg Feed				Contrast, *p* *		
	0	2.5	5	10	0 vs. DON	Linear	Quadratic
Femur weight (g)	11.2 ± 0.34	10.5 ± 0.34	10.4 ± 0.35	10.7 ± 0.34	0.105	0.359	0.141
Femur (% of leg weight)	7.0 ± 0.15	6.9 ± 0.15	7.1 ± 0.16	7.1 ± 0.15	0.827	0.575	0.901
Femur volume (mL)	9.4 ± 0.29	8.8 ± 0.29	8.9 ± 0.30	9.0 ± 0.29	0.109	0.311	0.227
Femur density (g/mL)	1.19 ± 0.01	1.19 ± 0.01	1.18 ± 0.01	1.20 ± 0.01	0.853	0.374	0.114
Tibiotarsus weight (g)	14.5 ± 0.63	14.4 ± 0.63	14.0 ± 0.65	14.3 ± 0.63	0.714	0.764	0.698
Tibiotarsus (% of leg weight)	9.0 ± 0.32	9.4 ± 0.32	9.5 ± 0.33	9.4 ± 0.32	0.243	0.330	0.491
Tibiotarsus volume (mL)	12.2 ± 0.56	12.4 ± 0.56	11.8 ± 0.58	12.1 ± 0.56	0.849	0.731	0.881
Tibiotarsus density (g/mL)	1.19 ± 0.01	1.16 ± 0.01	1.18 ± 0.01	1.18 ± 0.01	0.110	0.856	0.046

^1^ Values are least square means (LSM) ± standard error of the mean (SEM); *n* = 19 in the 2.5 mg/kg DON, 18 in 5 mg/kg DON and 19 in 10 mg/kg DON groups for all parameters, and *n* = 20 for the 0 mg/kg group; * *p*-values for orthogonal contrasts to test linear and quadratic relationships between control feeding and increasing levels of DON, as well as the overall difference of 0 DON versus all DON groups (0 vs. DON).

**Table 4 toxins-11-00352-t004:** Femur and tibiotarsus gravimetric characteristics in 5-week old broiler chickens fed diets with increasing levels of deoxynivalenol (DON; 0, 2.5, 5, or 10 mg DON/kg diet). ^1^

Item	DON, mg/kg Feed				Contrast, -*p* *		
	0	2.5	5	10	0 vs. DON	Linear	Quadratic
Femur DM (%)	44.2 ± 0.48	44.3 ± 0.50	44.7 ± 0.51	44.1 ± 0.51	0.785	0.940	0.552
Femur AsC_(FM)_ (%)	16.2 ± 0.27	15.7 ± 0.28	15.8 ± 0.29	15.9 ± 0.28	0.228	0.510	0.359
Femur AsC_(DM)_ (%)	36.6 ± 0.40	35.4 ± 0.41	35.5 ± 0.42	35.9 ± 0.41	0.034	0.259	0.052
Femur Asd (g/mL)	0.18 ± 0.004	0.18 ± 0.004	0.18 ± 0.004	0.18 ± 0.004	0.819	0.636	0.386
Tibiotarsus DM (%)	44.5 ± 0.63	43.9 ± 0.64	44.7 ± 0.66	45.1 ± 0.64	0.923	0.357	0.448
Tibiotarsus AsC_(FM)_ (%)	17.0 ± 0.53	15.7 ± 0.54	16.3 ± 0.56	17.3 ± 0.54	0.379	0.535	0.040
Tibiotarsus AsC_(DM)_ (%)	38.1 ± 0.66	35.9 ± 0.68	36.4 ± 0.70	38.0 ± 0.68	0.077	0.963	0.005
Tibiotarsus Asd (g/mL)	0.19 ± 0.006	0.17 ± 0.006	0.18 ± 0.007	0.20 ± 0.006	0.290	0.444	0.021

^1^ Values are least square means (LSM) ± standard error of the mean (SEM); *n* = 19 in the 2.5 mg/kg DON, 18 in 5 mg/kg DON and 19 in 10 mg/kg DON groups for all parameters, and *n* = 20 for the remaining 0 mg/kg group; * *p*-values for orthogonal contrasts to test linear and quadratic relationships between control feeding and the three increasing levels of DON, as well as the overall difference of 0 DON versus all DON groups (0 vs. DON). DM = Dry matter in percentage; AsC_(FM)_ = ash content in percentage related to the fresh matter; AsC_(DM)_ = ash content in percentage related to the dry matter; Asd = ash density in g/mL.

**Table 5 toxins-11-00352-t005:** Concentrations of Ca and P in serum, femur, and tibiotarsus, and Ca:P ratio of femur and tibiotarsus in 5-week old broiler chickens fed diets with increasing levels of deoxynivalenol (DON; 0, 2.5, 5, or 10 mg DON/kg diet). ^1^

Item	DON, mg/kg Feed				Contrast, -*p* *		
	0	2.5	5	10	0 vs. DON	Linear	Quadratic
Serum Ca (mmol/L)	2.65 ± 0.03	2.63 ± 0.03	2.57 ± 0.03	2.62 ± 0.03	0.215	0.244	0.291
Serum P (mmol/L)	2.45 ± 0.04	2.41 ± 0.04	2.35 ± 0.04	2.34 ± 0.04	0.073	0.034	0.716
Femur Ca (g/kg)	366 ± 1.37	364 ± 1.41	359 ± 1.44	360 ± 1.41	0.002	<0.001	0.415
Femur P (g/kg)	178 ± 0.69	178 ± 0.71	178 ± 0.73	179 ± 0.71	0.515	0.406	0.819
Femur Ca:P ratio	2.06 ± 0.009	2.05 ± 0.010	2.02 ± 0.010	2.01 ± 0.010	0.002	<0.001	0.639
Tibiotarsus Ca (g/kg)	364 ± 2.82	362 ± 2.90	350 ± 2.97	351 ± 2.90	0.003	<0.001	0.564
Tibiotarsus P (g/kg)	179 ± 0.82	180 ± 0.84	180 ± 0.86	180 ± 0.84	0.245	0.203	0.673
Tibiotarsus Ca:P ratio	2.04 ± 0.019	2.02 ± 0.019	1.94 ± 0.020	1.95 ± 0.019	0.003	<0.001	0.512

^1^ Values are least square means (LSM) ± standard error of the mean (SEM). Observation *n* = 19 in the 2.5 mg/kg DON, 18 in 5 mg/kg DON, and 19 in 10 mg/kg DON groups for all parameters, and *n* = 20 for the remaining 0 mg/kg group; * *p*-values for orthogonal contrasts to test linear and quadratic relationships between control feeding and increasing levels of DON, as well as the overall difference of 0 DON versus all DON groups (0 vs. DON).

**Table 6 toxins-11-00352-t006:** Pearson’s Correlation data between body weight (BW), daily DMI, Ca, P, and DON intake, bone parameters, and serum Ca and P in chickens fed diets with increasing levels of deoxynivalenol (DON; 0, 2.5, 5, or 10mg DON/kg diet).

Item	Leg Weight	Femur Weight	Femur Volume	Tibiotarsus Weight	Tibiotarsus Volume	Femur Ca	Femur P	Tibiotarsus Ca	Tibiotarsus P	Serum Ca	Serum P
**Body weight**	0.91 **	0.70 **	0.65 *	0.60 *	0.58 *	0.40 *	0.02	−0.21	0.18	0.28	0.31 *
**Daily DMI**	0.56 *	0.39 *	0.36 *	0.20	0.18	0.38 *	0.04	0.01	0.20	0.24	0.30 *
**Daily Ca intake**	0.57 *	0.39 *	0.36 *	0.20	0.19	0.38 *	0.04	0.01	0.20	0.24	0.30 *
**Daily P intake**	0.57 *	0.40 *	0.36 *	0.21	0.19	0.38 *	0.04	0.01	0.19	0.24	0.29
**Daily DON intake**	−0.08	−0.02	−0.04	0.00	−0.02	−0.35 *	0.12	−0.42 *	0.18	−0.07	−0.21

Note: * |*r*| ≥ 0.3; *p* < 0.05; ** |*r*| ≥ 0.7; *p* < 0.001; DMI, dry matter intake; DON, deoxynivalenol; Ca, calcium; P, phosphorus.

**Table 7 toxins-11-00352-t007:** Analyzed chemical composition of the basal diet. ^1^

Dietary Composition of 1 kg Feed	
Calculated metabolizable energy ^1^	12.66 MJ
Dry matter ^1^	889 g
Crude protein ^1^	207 g
Crude fat ^1^	78 g
Crude fiber ^1^	30 g
Crude ash ^1^	73 g
Starch ^1^	365 g
Sugar ^1^	52 g
Ca	14.6 g
P	7.3 g

^1^ Chemical composition is already published in Lucke et al. [7].

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
