# Peer review of "Deoxynivalenol in the Diet Impairs Bone Mineralization in Broiler Chickens"

_toxins, 2019, doi:10.3390/toxins11060352_

Round 1

Reviewer 1 Report

The amendments made by the authors are well received.

Reviewer 2 Report

The manuscript has improved in comparison to the original submission. I recommend its publication in the present form.

This manuscript is a resubmission of an earlier submission. The following is a list of the peer review reports and author responses from that submission.

Round 1

Reviewer 1 Report

Dear Authors,
The manuscript concerns the very important problem of the presence of deoxynivalenol in feed and the effect of this mycotoxin on the health of broiler chickens. However, the way the results are presented is very unfriendly to the reader and creates some substantive doubts.
In the introduction, the 
Authors suggest that a large number of locomotion problems is associated with the presence of mycotoxins, and in fact the vast majority of these problems is related to other causes. The Authors often refer to their previous manuscripts describing other results of the research. It makes reading and understanding the manuscript very difficult.
Table 1
is completely unclear. Presenting the value of meat:leg ratio is without scientific value.
The presentation of "pooled standard error of the mean" is also not appropriate. Generally, the applied statistical methods have been incorrectly selected for the conducted research.
The number of animals in the research groups differs in each tab, which is unacceptable. In toxicity studies, it seems appropriate to present the dose of toxin used per kg body weight. 
The presence of deoxynivalenol causes a decrease in feed intake!
In summary, the manuscript contains some scientific values, but the way they are presented is unacceptable.

Reviewer 2 Report

The article describes the effect of dietary DON exposure on bone characteristics and calcium and phosphorous levels in bone tissue of broiler chicks. The overall aim of the article is to identify the potential effect of DON and mobility and leg problems in chickens, which is an animal welfare concern.  The article is well written and presented.

However, one of the major concerns is related to the fact, that the manuscript presented here, is a second manuscript of an earlier published study (ref 7). Subsequently in the current manuscript, essential information for example on the total diet, including Calcium and Phototonus intake is lacking. Moreover, no data on feed intake (DON exposure and minerals) and growth rate of the animals is included. Looking back to the previous manuscript (ref 7), one of the prominent findings is a reduced feed intake and a reduced growth rate of the DON exposed animals. In this earlier publication, the authors clearly concluded that “the lower dry matter intake results directly in a reduced body weight of the chickens”. What is the evidence that the measured changes in bone weight and mineral content and bone composition are NOT related also to such and reduced feed (and mineral) intake? It seems therefor essential to add these previous findings to the current manuscript and address these aspects in the discussion.  Moreover, the current presentation of results contains only rather simple gravimetric measurements and Ca and P concentrations. To identify specific (toxicological) effect exerted by DON, functional bone-related biomarkers (such as osteocalcin) should be added and more refined structural analyses of the bone tissue (histo-morphometry, Saframin/PSR staining) should be added. All these parameters/biomarkers are well described for poultry (many in the context of efficacy testing of phytases). The authors are therefore invited to include such additional data.

Minor editorial comments:

Pleas add in the abstract DON mg/kg feed (the word feed is needed although dietary exposure is mentioned in the same sentence).

The reference to DTP-1 should be critically evaluated, as this refer to earlier case-studies with non-defined (water soluble) extracts of mouldy feed. The nature of this toxin has not been elucidated yet.  

Tables: the statistical presentation is rather unusual for a biomedical manuscript. At least a more transparent presentation of the findings (means and SD or SEM per individual group) should be presented in a supplementary table.

Please also check the reference list (which is quite long) for consistency. For example, refs 3,5,6,17,31,41 seem to be only a marginal interest here and should be replaced by references addressing the various aspects of bone metabolism.

Reviewer 3 Report

The manuscript is scientifically sound, the data are clearly presented and documented. However, the manuscript is just a follow-up of a paper Lucke et al., Mycotoxin Res. 2017 and I am not sure if its significance is sufficient for a separate publication. Although leg problems in chicken are common and result in huge economic losses, it is rather unlikely that mycotoxins are an important factor, taking into account their levels in feed. Even though the authors show the statistically significant changes in the tested parameters, these could be as well coming from the overall effect of DON on the nutrition efficiency.  Hence, these results would perfectly suit the mentioned paper relating to the basic health and production status of birds.

I have also two minor remarks regarding the interpretation of data, discussion and drawing the conclusions.

Based on the correlation matrix (Figure 1), the factor the mostly correlated with the tested parameters was body weight. In my opinion, there was actually no proof that DON specifically impaired the mineralisation (I mean with the mechanism other than just lower elements absorption from GI tract). Did the authors think about correlating BW with Ca and P plasma levels? In my opinion, it could confirm/ overthrow the hypothesis that the observed results came from lower nutrition efficiency. 

I find it difficult to follow the discussion on the differences between the femur and tibiotarsus. First, the discussed results are only in the text and it requires some effort to find what the authors relate to. Second, is there some scientific data that could confirm the hypothesis drawn in lines 174-178? It seems rather odd to me.